# Evaluation of Selected MRI Parameters in the Differentiation of Mucinous Ovarian Carcinomas and Metastatic Ovarian Tumors

**DOI:** 10.3390/cancers16213569

**Published:** 2024-10-23

**Authors:** Marta Halaburda-Rola, Laretta Grabowska-Derlatka, Leszek Kraj, Rafal Stec, Pawel Derlatka

**Affiliations:** 1Second Department of Clinical Radiology, Medical University of Warsaw, 02-091 Warszawa, Poland; marta.haburda-rola@wum.edu.pl; 2Department of Oncology, Medical University of Warsaw, 02-091 Warszawa, Poland; leszek.kraj@wum.edu.pl (L.K.); rafal.stec@wum.edu.pl (R.S.); 3Department of Molecular Biology, Institute of Genetics and Animal Biotechnology, Polish Academy of Science, 01-447 Magdalenka, Poland; 4Second Department of Obstetrics and Gynecology, Medical University of Warsaw, 02-091 Warszawa, Poland; pawel.derlatka@wum.edu.pl

**Keywords:** MRI, mucinous ovarian carcinoma, metastatic ovarian tumor

## Abstract

Initial evaluation of ovarian lesions is essential in proper preoperative assessment. Correct preoperative differentiation between mucinous ovarian carcinomas and metastatic ovarian tumors enables precise treatment planning and the individualization of therapy. The correlation of selected MRI parameters enables precise differential diagnosis.

## 1. Introduction

Approximately 15–30% of malignant tumors in the ovary are metastatic lesions [1,2]. They most often come from the colon (30%), stomach (16%), breast (13%), pancreas (12%), and biliary tract (15%). Among gynecological cancers, ovarian metastasis arises from endometrial cancer (23%) [3,4].

Metastatic ovarian tumors (MOTs) originating in the gastrointestinal tract are most often diagnosed based on a history of colorectal cancer or gastric cancer, as well as the bilateral presence of lesions. However, MOTs and primary ovarian cancer (POC) often show morphologic similarity on computed tomography (CT) and magnetic resonance imaging (MRI) and may have a similar clinical course. Diagnostic difficulties arise especially in situations where an ovarian tumor is the first sign of the disease. Misdiagnosis leads to inappropriate qualification for surgical treatment. There are particular diagnostic difficulties in differentiating MOTs and primary mucinous ovarian carcinomas (MOCs) [5]. The scopes of surgery for POC versus MOTs differ substantially. Adjuvant chemotherapy regimens are also different [6,7,8].

Previous publications have found it difficult to differentiate POC (particularly MOCs) and MOTs originating in the gastrointestinal tract based on morphological criteria on CT and MRI. The morphological features of MOCs and MOTs are very close or even identical [2,8,9,10].

It is also possible to differentiate MOTs and POC using tumor markers. The most common tumor antigens used in differential diagnosis are CA 125, CA 19-9, and cancer-embryonic antigen (CEA) [8,11,12]. Morphologic classifications combined with marker findings have also been created to improve diagnostic capabilities, such as the “mille-feuille sign” described on MRI, as well as CA 125 and CEA. Such an image is supposed to be indicative of tumor metastasis from the colon to the ovary [8]. Seidman et al. proposed a very simple way to differentiate MOCs and MOTs based on tumor size and unilateral or bilateral occurrence. Primary ovarian origin is supposed to be evidenced by unilateral occurrence and a diameter greater than 10 cm. Such coincidence is supposed to give an approximately 90% probability of diagnosing MOTs [13]. Of course, these are very approximate criteria. In modern MR imaging of POC types, not only morphological criteria are relevant. Diffusion (DWI) and perfusion (DCE) parameters play an important role. The use of ADC (apparent diffusion coefficient) maps or determination of time to peek (TTP), perfusion maximum enhancement (Perf Max En) or volume transfer constant (Ktrans) values in DCE imaging already allows us to differentiate between low-grade serous ovarian cancer (LGSOC), high-grade ovarian cancer (HGSOC), and mucinous ovarian carcinoma (MOC) [14,15,16,17,18].

The purpose of this study was to evaluate the usefulness of diffusion and perfusion parameters on MR imaging for differentiating MOTs originating from the gastrointestinal tract and primary MOCs and comparative analysis with existing morphological criteria.

## 2. Materials and Methods

### 2.1. Study Protocol

The study was conducted prospectively in the 2nd Department of Clinical Radiology in the cooperation with the 2nd Department of Obstetrics and Gynecology in the Medical University of Warsaw. The inclusi on criterion for the study purposes was a unilateral or bilateral, solid mass or mixed solid/cystic mass, diagnosed in CT or a transvaginal ultrasound examination. Patients with a coexisting second cancer, with contraindications to MRI with gadolinium contrast examination, or those under 18 were excluded from the study.

2014 WHO criteria were applied for the determination of the type and histological grade of cancer. Patients with histopathological confirmation of primary MOCs and metastatic lesions from gastrointestinal tract were included into the study. The following immunohistochemical (IHC) profile was applied: CK7, CK20, PAX8, CDX2 WT1, MUC2, ER, and PR [19].

### 2.2. MRI Protocol

Preoperative MRI procedures were performed with a 1.5T MR scanner (MAGNETOM Avanto, Siemens AG, Erlangen, Germany).

The MRI protocol contained the following sequences: turbo spin-echo (TSE), T2-weighted images (T2WI), fat suppressed T2-weighted images (fsT2WI), turbo inversion recovery magnitude (TIRM), diffusion-weighted echoplanar imaging (DW-EPI), and pre- and postcontrast dynamic T1-weighted gradient echo sequences (3D T1-GRE). The detailed MR study protocol is presented in Table 1.

Multi-slice EPI unified images were used for an axial DW imaging sequence with 30 × 6 mm slices (pelvic part) 360 × 360 mm FoV; TR = 4250 ms; TE = 73 ms; with diffusion weights of 0, 50, 500, 1000, and 1500 mm^2^/s. The parameters are collected in Table 2. Motion correction was completed automatically.

MRI scans were assessed by two board-certified oncologic imaging radiologists experienced in the analysis of pelvis MRI.

For morphological analysis, Seidman’s criteria were used, dividing ovarian tumors according to the following criteria: unilateral or bilateral occurrence and tumor diameter < 10 cm and ≥10 cm [13]. The incidence of the “mille-feuille” sign has also been described [8]. Regions of interest (ROI) were drawn on the apparent diffusion coefficient (ADC) maps, and all b values DWI were outlined in the Multimodality Workplace Station (GE AW Serwer 3.2 ext. 4.0, Volume Viewer16.0 Ext. 2 Ready View).

Each examination was analyzed twice by each radiologist preoperatively and separately. The only information they had at the time of the MRI analysis was the presence of a suspicious pelvic mass and the results of marker tests. To compare the ADC values recorded by the two radiologists and for the purpose of analyzing these values, the following markings were adopted, whereas ADC1 and ADC2 correspond to the analysis performed by the first radiologist, while ADC3 and ADC4 correspond to the analysis performed by the second radiologist.

In the DWI sequence (with b values of 0, 50, 500, 1000, 1500 mm^2^/s), the ROI contained a circle with a diameter of 5 mm, which was placed in the solid part of the tumor, avoiding areas of necrosis, partial volume effects, and artifacts. ROIs were copied from the DWI to the corresponding ADC maps, and the ADC values were recorded. The dynamic contrast enhancement (DCE) sequence as well as T1WI (non-contrast and post-contrast) and the parameters for dynamic analysis are presented in Table 1.

ROIs were marked on the post-contrast DCE images and duplicated to DCE parametric maps. During DCE image acquisition, non-contrast images were acquired first, followed by contrast agent administration and continued acquisition. On the DCE images, the following parameters were measured: perfusion maximum enhancement (Perf. Max En.) and time to peek (TTP). DCE parametric maps were generated automatically using Workplace Station. The contrast agent applied in all patients was gadobutrol (Gadovist, Bayer AG, Reading, UK) administered as a bolus dose of 0.1 mmol/kg, followed by a bolus dose of 20 mL of physiological saline (NaCl 0.9%).

### 2.3. Statistical Analysis

IBM SPSS Statistics (version: 29.0.1.0(171)) was used to analyze the distribution of variables, perform some statistical tests, as well as calculate some statistics. PQStat (version: 1.8.6.102) was used for the same purpose to analyze the data and to prepare all the visualizations seen in this article. Power BI (version: 2.120.731.0) was used to create charts for the qualitative criteria of the studied tumors.

The Shapiro–Wilk test was used to examine if a variable is normally distributed. In the calculated statistics, all the numerical values which did not have a normal distribution were presented as the median and the first and the third quartiles (Q_1_–Q_3_). All numerical values characterized by a normal distribution are presented as means and standard deviations, while qualitative variables are presented as raw numbers and percentages. 

To determine the *p*-value, three different tests were used. The main aim was to define if the *p*-value is less than 0.05, what indicates a statistically significant difference. For interval scale variables that do not meet the condition of normal distribution and for those that are independent, the Mann–Whitney U test was used, which is specifically designed for such conditions. For quantitative variables that meet the condition of normal distribution, the t-test for independent groups was used. For nominal and independent variables, the chi-square/Fisher’s exact test was used. The used statistical tests applied to independent samples were designed to compare the mean values (or to compare the values in which a parameter was measured) of two independent groups to see if the associated means (or different measurements) of the test samples differed statistically significantly.

To determine the interobserver agreement between the two reviewers, two tests were used. To check the significance of the nominal variables the test of kappa coefficient was applied. For the interval variables, a test to check the significance of the ICC coefficient (intraclass correlation coefficient) was used.

## 3. Results

In the performed analysis, 35 patients were examined; to achieve the aim of the study, they were divided into 2 independent groups. The first group consisted of patients with primary ovarian cancer, whereas the second group consisted of patients with a diagnosed metastatic ovarian tumor. 

The mean age of the patients in the first group was 49 years (*SD*: 12.1), and for the second group the age was almost 59 years (*SD*: 11.5, *p*-value = 0.04).

The results, presented as the mean with standard deviation in Table 2, confirm other published data which say that primary ovarian cancer occurs among younger patients.

A statistically significant difference in the age of the patients is shown graphically using a box plot (Figure 1).

In a next step of the study, the tumors present among the patients studied were analyzed. In the group of examined patients, 22 (63%) patients suffered from bilateral (two-sided) tumors, and, thus, 13 patients (37%) possessed unilateral (one-sided) tumors. On that basis, 57 tumors were analyzed, which were also divided into 2 groups, i.e., the first group (primary mucinous ovarian cancer) and the second group (metastatic ovarian tumors).

To analyze the characteristics of the 57 tumors, we examined whether they were characterized by being bilateral (one-sided/two-sided) and whether they met the criterion of the 10 cm limit, and assigned them to the appropriate Seidman categories, as described in Table 3.

The results of this analysis are shown in Table 4.

Based on the obtained *p*-value, it can be concluded that statistically significant differences were recorded for all three tested parameters. The obtained results are presented graphically as pie charts (Figure 2) and as stacked column charts (Figure 3 and Figure 4).

Selected parameters for 57 tumors were analyzed, including apparent diffusion coefficients (ADC), time to peek (TTP), and perfusion maximum enhancement (Perf. Max. En.) divided into two groups, as defined above.


**ADC parameter—apparent diffusion coefficients**


For the apparent diffusion coefficients parameter, the data are shown in Table 5.

Based on the results in Table 5, it can be concluded that the median for parameters ADC1, ADC2, ADC3, and ADC4 is statistically higher in the group of patients suffering from POC than in the group of patients diagnosed with a metastatic ovarian tumor. The differences for the apparent diffusion coefficients parameter between the two groups are shown graphically in the form of box plots (Figure 5, Figure 6, Figure 7 and Figure 8).


**TTP and Perf. Max. En.**


The collected data for 57 tumors were also examined for 2 other parameters, i.e., time to peek (TTP) and perfusion maximum enhancement (Perf. Max. En.). The results of this analysis are shown in Table 6.

Based on the results in Table 6, it can be concluded that the statistical difference only relates to the TTP parameter. The median value of the TTP parameter is statistically significantly higher in the group of patients diagnosed with primary ovarian cancer (median TTP: 410; Q_1_–Q_3_: 370–465) compared to the median value of TTP in the group of patients diagnosed with a metastatic ovarian tumor (median TTP: 154; Q_1_–Q_3_: 147.5–161, *p*-value = 0.0001). This result is shown graphically in a box plot (Figure 9).


**Interobserver agreement**


For each scoring system, 114 observations were submitted.

There was statistically significant interobserver agreement between the two observers in assessing qualitative tumor using the mille-feuille sign. The interobserver concordance oscillated at the level of good agreement (*kappa* value = 0.64) for two categories, i.e., the absence and the occurrence of the mille-feuille sign (Table 7).

As part of the interobserver agreement analysis, an error plot was determined for the two observers (Figure 10), which graphically shows the same values as shown in Table 7.

The mille-feuille sign was noted in 30% patients with MOCs and 64% patients with MOTs.

In addition, an agreement plot (Figure 11) was created, which shows how each tumor from the first to the fifty-seventh was rated by the two observers. The columns on the chart marked by two colors represent tumors that doctors evaluated differently. The same color column in the chart means 100% agreement between doctors on the mille-feuille sign aspect.

Intraclass correlation (ICC) was used to examine interobserver agreement for the quantitative method, i.e., the ADC parameter. Based on the ICC values, good interobserver reliability was found for the method, where ICC = 0.84 (Table 8).

To graphically represent the interobserver agreement between the two observers in the quantitative method shown in Table 8, a multiple dot graph was created (Figure 12).

Table 9 summarizes the results of the analysis of the ROC curves. Figure 13, Figure 14, Figure 15, Figure 16 and Figure 17 present the ROC curve analysis for the selected parameters separately.

Selected images depicting analyzed MOTs and MOCs are presented in Figure 18 and Figure 19, respectively.

## 4. Discussion

The presented work demonstrates new possibilities for preoperative differentiation of MOCs and MOTs using DWI and DCE MRI. Previously, the differentiation of MOCs and MOTs based on morphological criteria was not very precise. Considered criteria included tumor size, unilateral or bilateral location, and the presence of solid elements [4,8,13,20].

Our study confirmed the usefulness of morphological criteria, but most importantly, we demonstrated the great value of quantitative criteria based on DWI and DCE parameters in differentiating between MOCs and MOTs. We showed statistically significantly higher ADC values for MOCs than for MOTs in four consecutive measurements (1465.5 vs. 842, 1434 vs. 821, 1631.75 vs. 842, and 1442 vs. 845, respectively; in each measurement *p* = 0.0001) Similar differences (higher values for MOCs than for MOTs) were shown by the TTP parameter in the DCE study (410 vs. 154; *p* = 0.0001).

Mucin-producing ovarian tumors can be primary epithelial neoplasms or metastases to the ovary from adenocarcinoma, usually originating in the gastrointestinal tract. Ovarian metastases are uncommon, occurring in 2.1–13.6% of patients following colorectal resection, and occur usually in the context of disseminated peritoneal disease [21]. Colorectal cancer metastases with mucinous histology mimic primary MOCs on imaging studies. On MR imaging, mucinous tumors of any origin may contain multicellular cystic areas with a “stained-glass” appearance on T2WI, as well as thickened septa and small solid areas [2,4,22,23,24].

Many authors emphasize that the age of patients can also be a differentiating factor. MOCs occur in younger patients than MOTs [24,25]. In our material, patients with MOCs were younger than those with MOTs, although the difference was on the verge of statistical significance (49.3 ± 12.1 vs. 58.7 ± 11.5; *p* = 0.004).

Bilateral MOT lesions are clearly more common than among MOCs. This is one of the criteria developed by Seidman, used in differential diagnosis. Stomach or colon metastases typically appear as complex masses in both ovaries. They are usually solid, but cystic components may also be present [2,24,25]. Bilaterality is the key imaging finding, as seen in other secondary tumors of the ovaries. The second criterion is the diameter of the tumors. It divides lesions above and below 10 cm. The combination of unilateral or bilateral positioning and a diameter < 10 ≤ cm may indicate a primary or metastatic origin of the ovarian tumor. The authors believe that for lesions occurring bilaterally with a diameter < 10 cm and ≥10 cm, the tumor is 90% metastatic. On the other hand, for lesions ≥ 10 cm and with a unilateral occurrence, the probability of an MOC is 92% [13]. A similar study was published by Song-Qi et al., where 35.9% of the patients with MOTs were discovered synchronously with the primary carcinomas, 25.6% of the patients with MOTs were bilateral, and all patients with MOCs had unilateral lesions [26].

In our material, we demonstrated the concordance of Seidman’s criteria in differentiating MOCs and MOTs at 82% for unilateral lesions ≥ 10 cm among MOCs (*p* < 0.000001). The probability of diagnosing MOTs for unilateral tumors < 10 cm was 87%, the probability for bilateral tumors ≥ 10 cm was 95%, and the probability for bilateral tumors < 10 cm was 92% (*p* < 0.000001).

Yemelyanova et al. tried to improve the efficiency of the method and relate it to the starting point of the metastatic lesion. By adjusting the size criterion to 12 cm, the efficiency of the algorithm was maintained for MOCs and improved for MOTs, with correct diagnosis of a total of 86% of cancers, including 100% of MOCs and 80% of MOTs. Sensitivity was optimized at 13 cm, with correct classification of a total of 87% of cancers, including 98% MOCs and 82% MOTs (colorectal, 80%; appendix, 79% of low-grade tumors; pancreatic and biliary cancers, 100%; small intestine, 33%, stomach, 100%; cervix, 70%). The largest number of exceptions were metastatic colorectal and cervical cancers, despite using the optimized size criterion. Colorectal metastasis was found to be the most common form of metastasis, but also the one that broke out of the algorithm the most [27].

Hu et al. also moved the cutoff point for differentiating MOCs and MOTs to a tumor diameter of 13 cm. They showed that a tumor size of <13 cm for MOTs had a sensitivity of 80% and a specificity of 80%. The most common exception to the <13 cm cutoff size for metastasis also turned out to be colorectal cancer, of which 30% were ≥13 cm in size. The algorithm, in which a tumor ≥ 13 cm is considered primary if it shows no surface nodules or is bilateral, and a tumor < 13 cm is considered metastatic if it is not unilateral, correctly classified 94% (64/68) of metastatic tumors and 98% (60/61) of primary tumors [28].

Kurokawa et al. pointed out another important morphological feature of mucinous tumors in differentiating primary and secondary lesions from colorectal cancer, described as the ‘mille-fueille sign’ on both CT and MRI scans. The name “mille-feuille” comes from the French pastry of the same name, in which the pastry and cream are arranged alternately, and refers to fine layered structures with layers a few millimeters apart and a width/length ≥ 10/20 mm, respectively [8]. In this study, morphological lesions were divided into four groups: with a mille-feuille sign, solid and cystic, multicystic without nodules, and multicystic with nodules. Histopathologically, primary ovarian cancers included cases of HGSOC, endometrioid ovarian cancer, clear cell carcinoma, MOCs, seromucinous carcinoma, and malignant Brenner tumors. The mille-feuille sign was observed more frequently in colorectal MOTs than in POC (8/41–19.5% vs. 1/46–2.8%, *p* = 0.01; the PPV (positive predictive value) and NPV (negative predictive value) were 0.89 vs. 0.58, respectively). Morphologic images in the form of multicystic without nodules were more common in colorectal MOTs (*p* = 0.041) while solid and cystic images were more common in POC (*p* < 0.001). There were no differences in the incidence of the morphologic form of multicystic with nodules between POC and colorectal MOTs (*p* = 0.14) [8].

In our material, in which we considered only MOCs among primary lesions, we showed the presence of a mille-feuille sign among MOCs and MOTs in 30% and 64% of cases, respectively.

There was high concordance among the radiologists describing the lesions in both finding the presence and absence of the mille-feuille sign. In both cases, the k value was 0.64 (95%CI 0.38–0.9; *p* = 0.000002).

There have also been attempts to differentiate the starting point of ovarian metastatic tumors based on the morphologic description of their features on CT scan, relating them to MRI images and to histopathologic diagnosis. On this basis, Kato et al. proposed four macroscopic types of lesions. Type one, oval, homogeneous-solid, was supposed to be indicative of metastases of gastric cancer to the ovary. Type two, heterogeneous-solid and small with a multinodular surface, were supposed to correspond to breast metastases. The third type, solid-cystic and predominantly solid lesions, were supposed to be indicative of endometrial cancer origin. The last, fourth type, cystic-solid and multilocular lesions with solid components, attested to primary ovarian cancer [29].

However, we believe that morphological features are not sufficient to determine the type and degree of malignancy of an ovarian tumor. Additional information can be provided by quantitative data on MRI imaging, such as DWI and DCE, on dynamic examination with a contrast agent. As emphasized by many authors describing primary ovarian tumors, these are important data in differential diagnosis. There have been publications, including from our center, attesting to the excellent capabilities of DWI and DCE MRI in differentiating specific types of POC [14,16,17,30,31,32]. DWI and DCE parameters have been shown to be compatible with immunohistochemical determinations, and their value in predicting response to POC treatment has been noted [15,16].

The literature to date lacks items showing the usefulness of DWI and DCE in differentiating between MOCs and MOTs.

In the paper, Xu et al. described one of the first such attempts. However, they did not measure quantitative diffusion and perfusion parameters. They only described that primary and metastatic lesions within solid parts showed features of moderate to severe diffusion restriction. Similarly, they performed dynamic studies after LAVA-DCE contrast, where they found moderate to strong enhancement of solid parts. Specific perfusion parameters were not measured. Hence, in their conclusion, the authors state that MOCs and MOTs are not statistically different in DWI and DCE imaging [33,34].

We showed clear differences in diffusion imaging between MOCs and MOTs. The two groups also differ in terms of perfusion parameters. Therefore, we believe that morphological parameters based on tumor size, unilateral or bilateral presence (Seidman criteria), and description of tumor structure or symptoms, such as ‘mille-feuille’, are only a preliminary diagnosis. After initial evaluation, it is necessary to determine the values of DWI and DCE parameters.

When comparing methods for differentiating MOC and metastatic tumors, the highest sensitivity and specificity were recorded for the ADC and TTP parameters. The former corresponds to the greater cellularity of metastatic tumors, while the second indicates their greater vascularization. Such high AUC values for TTP may obviously be due to the small size of the group, but certainly shows the usefulness of DCE in differentiating between MOCs and metastatic tumors. A similarly high AUC was shown by the Seidman criteria. The lowest value, although still with significant sensitivity and specificity, was shown for the mille-feuille sign (AUC 0.74). Perhaps this is related to the fact that the previously mentioned parameters are measurable, quantifiable (values of ADC, TTP, tumor diameter), or unambiguous in interpretation (unilateral or bilateral occurrence). In contrast, when assessing the mille-feuille sign, there is an element of unquantifiable subjective assessment.

The clinical implication of our study is the diagnostic model, which involves the initial application of Seidman’s criteria and then combining them with diffusion parameters (ADC maps) and basic perfusion elements (DCE), e.g., TTP and Perf. Max.En., in order to differentiate primary and metastatic ovarian tumors. Quantitative determination of DCE parameters in particular is not currently widely used.

Our work has several limitations. Firstly, the main limitation was the single-center nature of the study and, secondly, the small number of the patients enrolled to the study was another limitation.

However, narrowing the group to mucinous type allowed for standardization of the study group. Statistically clear differences in mucinous primary and secondary tumors that are morphologically very similar warrant further multicenter studies and confirmation of our preliminary diagnoses.

## 5. Conclusions

Morphological criteria in imaging studies have lower sensitivity and specificity in the differential diagnosis of ovarian tumors compared to quantitative criteria. Quantitative criteria, such as ADC, are a very good tool for differentiating primary mucinous ovarian carcinomas (MOCs) and metastatic tumors (MOT)s. Among the analyzed DCE parameters, very good results in differentiating MOCs and MOTs were obtained in the TTP. In the preoperative differential diagnosis of ovarian tumors, the use of morphological criteria according to Seidman, extended by the assessment of changes on ADC and TTP maps, seems to be a sufficient method of differentiation.

## Figures and Tables

**Figure 1 cancers-16-03569-f001:**
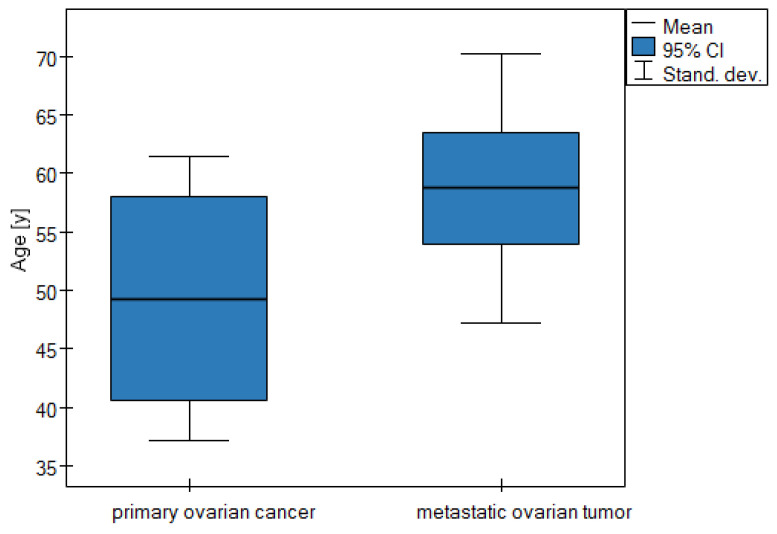
Box plot of mean age comparisons for the two groups of patients.

**Figure 2 cancers-16-03569-f002:**
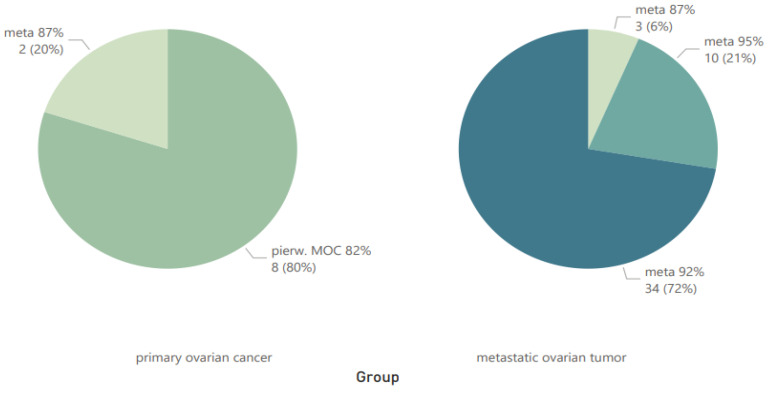
The Seidman criterion by group.

**Figure 3 cancers-16-03569-f003:**
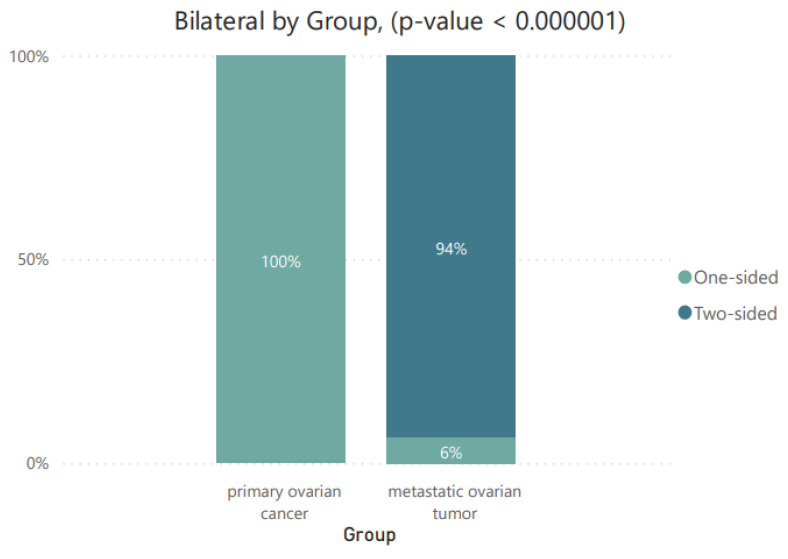
Bilateral criterion by group.

**Figure 4 cancers-16-03569-f004:**
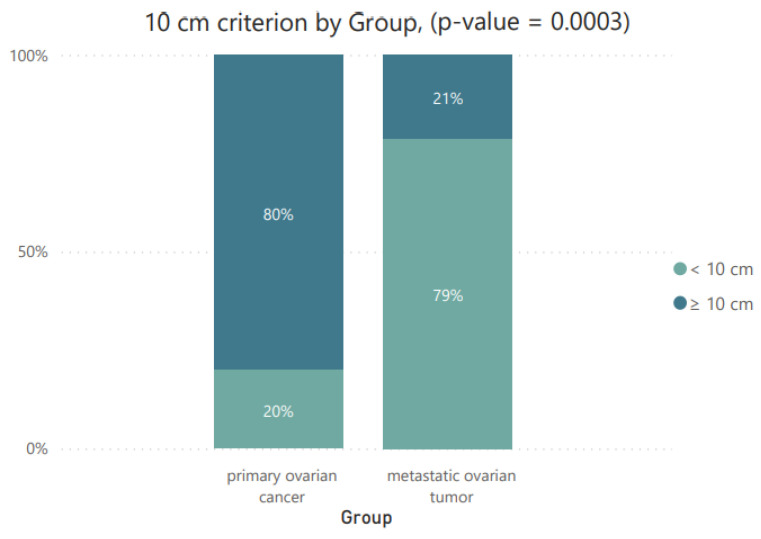
10 cm criterion by group.

**Figure 5 cancers-16-03569-f005:**
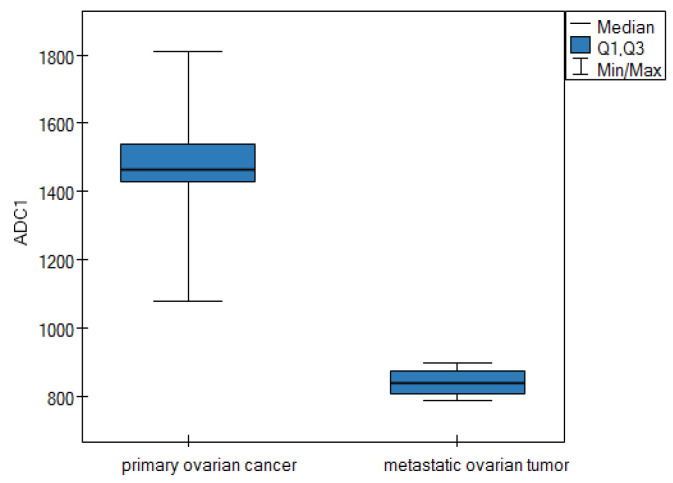
Box plot of comparisons of the median ADC1 parameter for two groups of patients.

**Figure 6 cancers-16-03569-f006:**
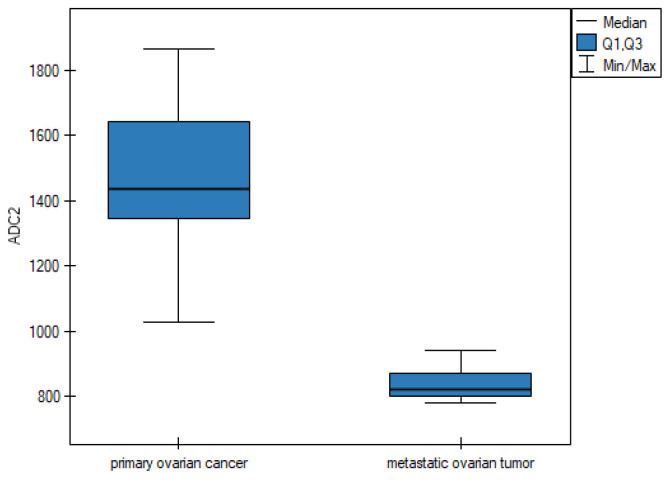
Box plot of comparisons of the median ADC2 parameter for two groups of patients.

**Figure 7 cancers-16-03569-f007:**
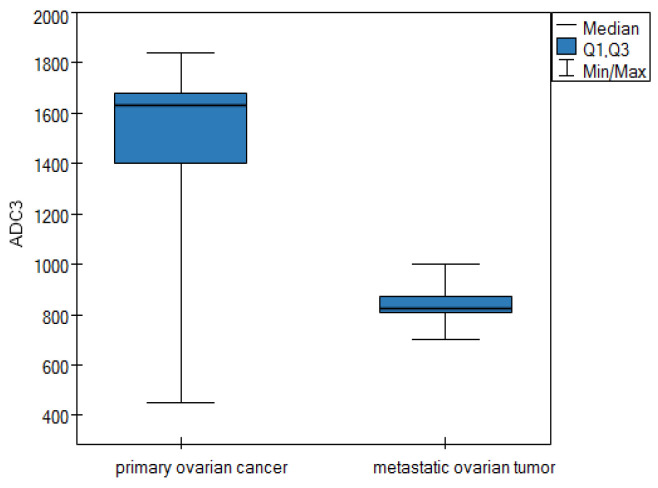
Box plot of comparisons of median ADC3 parameter for two groups of patients.

**Figure 8 cancers-16-03569-f008:**
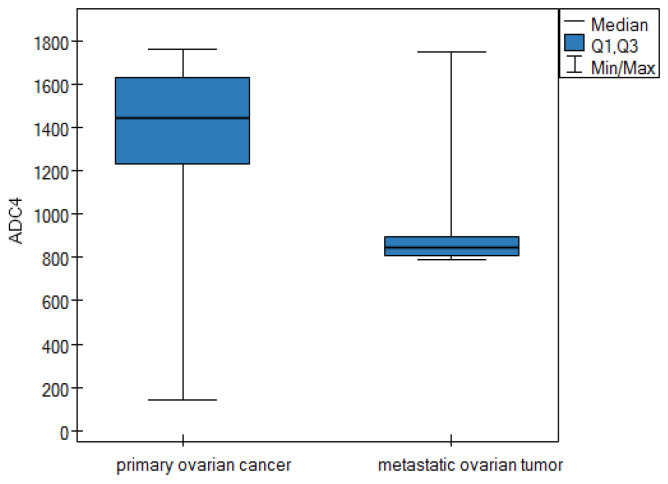
Box plot of comparisons of the median ADC4 parameter for two groups of patients.

**Figure 9 cancers-16-03569-f009:**
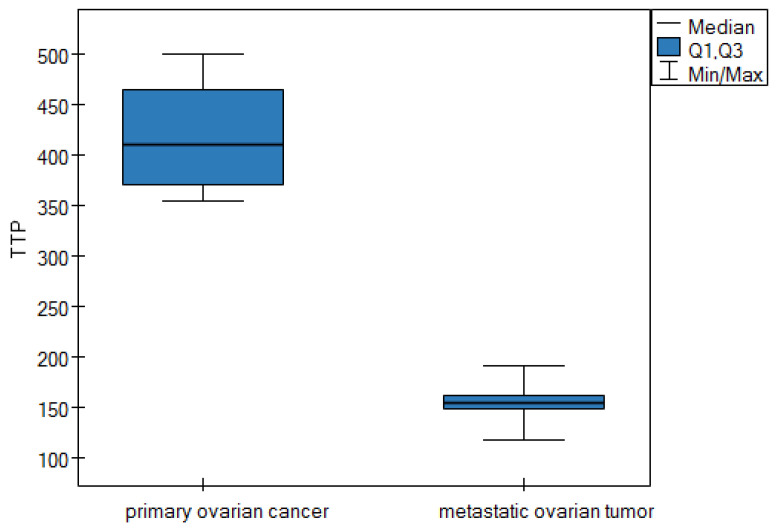
Box plot of comparisons of the median TTP parameter for two groups of patients.

**Figure 10 cancers-16-03569-f010:**
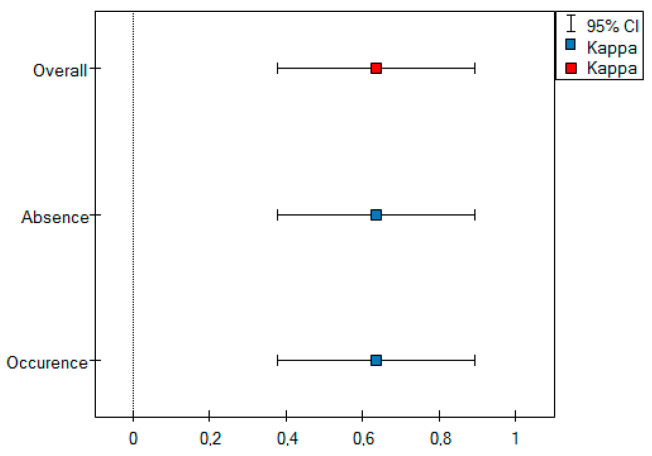
Error plot for the mille-feuille sign method.

**Figure 11 cancers-16-03569-f011:**
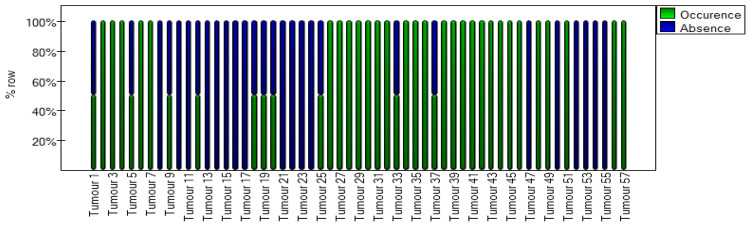
Agreement stacked column plot for the mille-feuille sign method.

**Figure 12 cancers-16-03569-f012:**
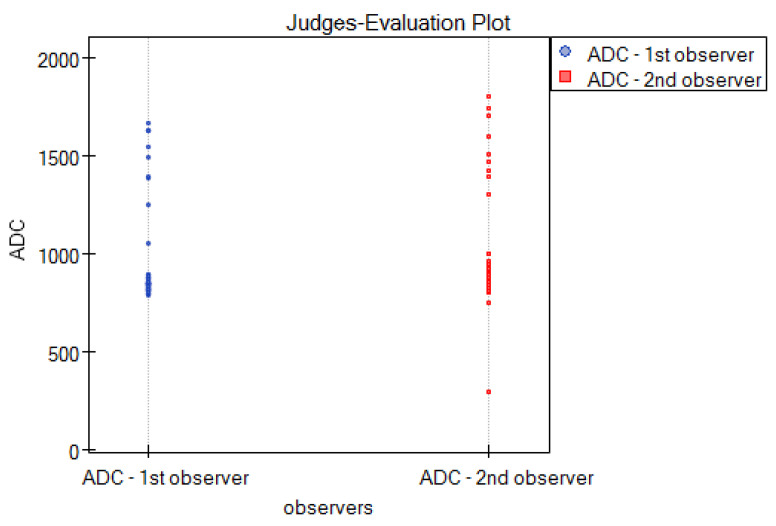
Multiple dot graphs for the interobserver agreement for the ADC scoring system.

**Figure 13 cancers-16-03569-f013:**
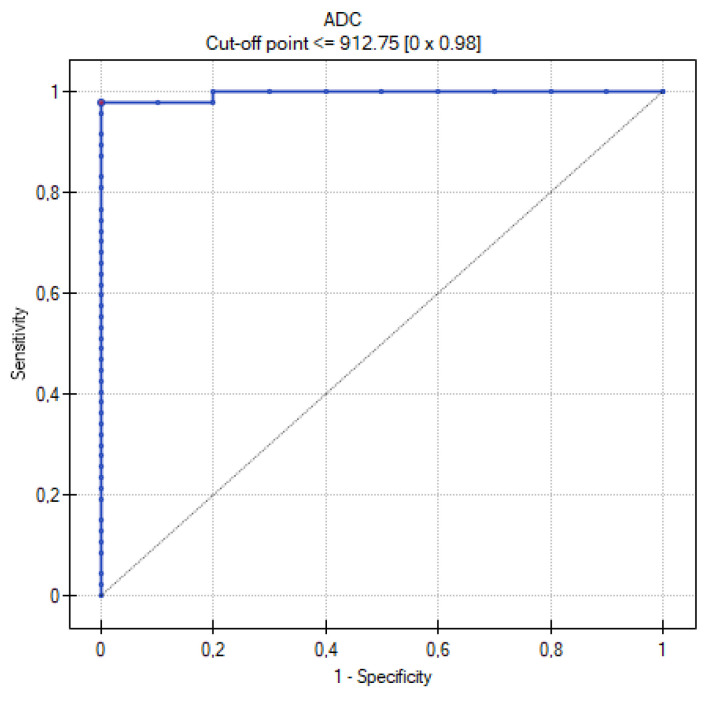
ROC curve for the diagnostic performance of the ADC level between primary and metastatic ovarian tumors.

**Figure 14 cancers-16-03569-f014:**
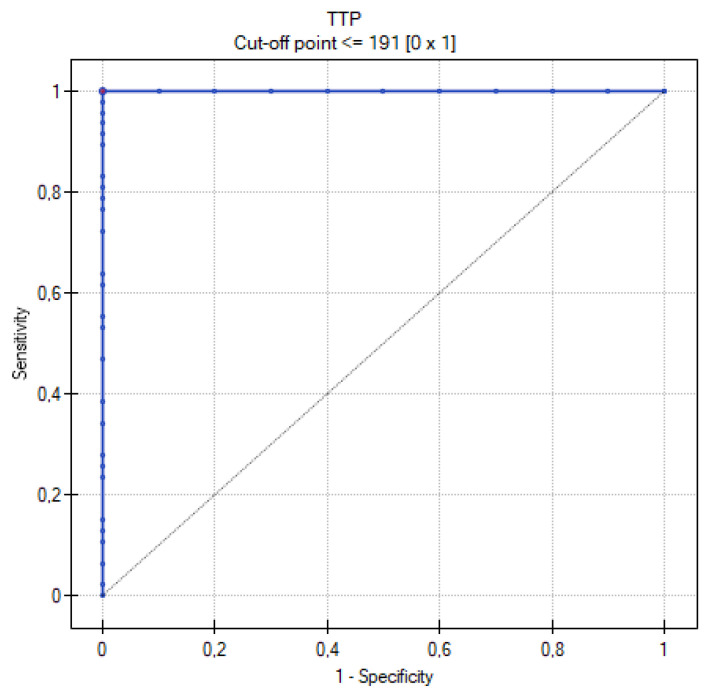
ROC curve for the diagnostic performance of the TTP level between primary and metastatic ovarian tumors.

**Figure 15 cancers-16-03569-f015:**
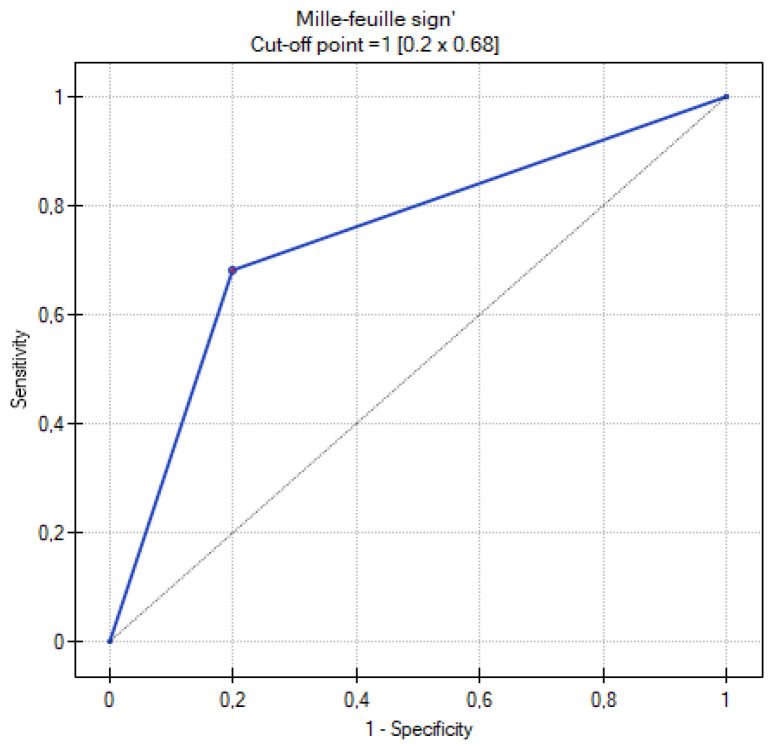
ROC curve for the diagnostic performance of the mille-feuille sign level between primary and metastatic ovarian tumors.

**Figure 16 cancers-16-03569-f016:**
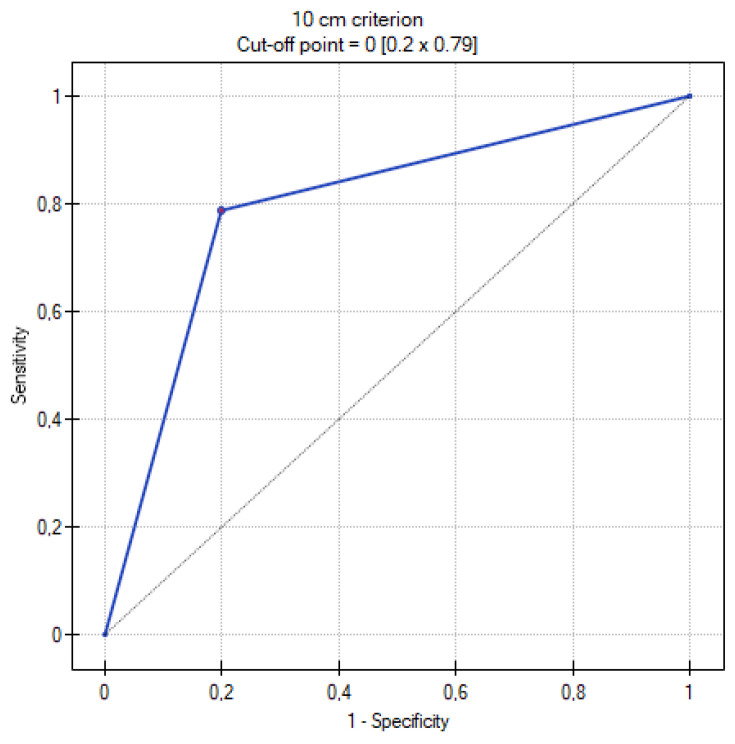
ROC curve for the diagnostic performance the of 10 cm criterion level between primary and metastatic ovarian tumors.

**Figure 17 cancers-16-03569-f017:**
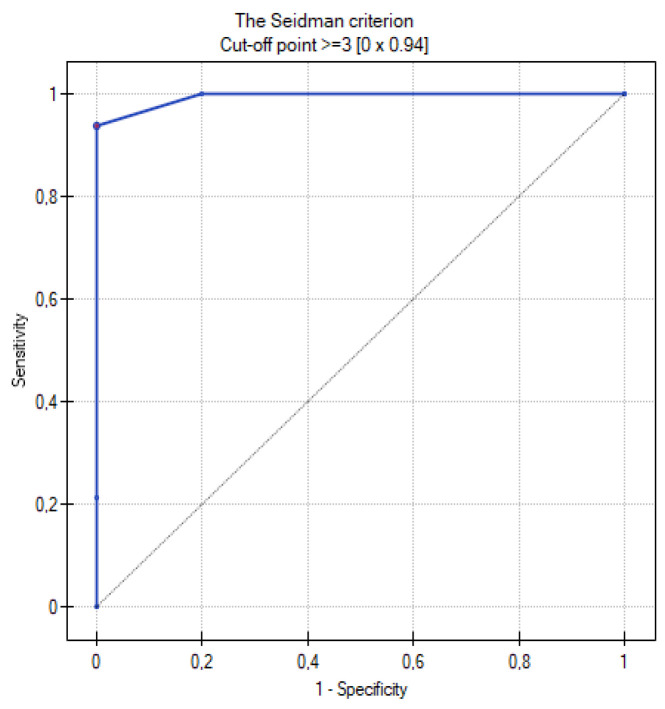
ROC curve for the diagnostic performance of the Seidman criterion level between primary and metastatic ovarian tumors.

**Figure 18 cancers-16-03569-f018:**
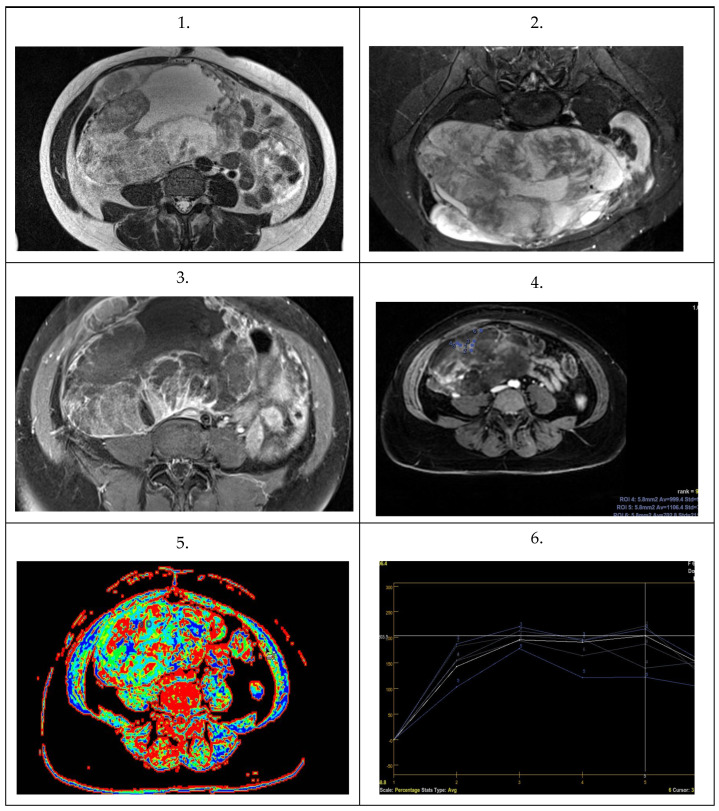
Images from 58–year-old woman with an MOT depicting a large, multi-cystic tumor with a striped structure compatible with a mille-feuille sign. (**1**) Axial T2–weighted image in the pelvis. (**2**) Axial fat–suppressed. (**3**) Axial T1–weighted post-contrast with fat suppression. (**4**) Diffusion ADC maps. (**5**,**6**) Perfusion maps and contrast enhancement curves from intratumoral solid tissue.

**Figure 19 cancers-16-03569-f019:**
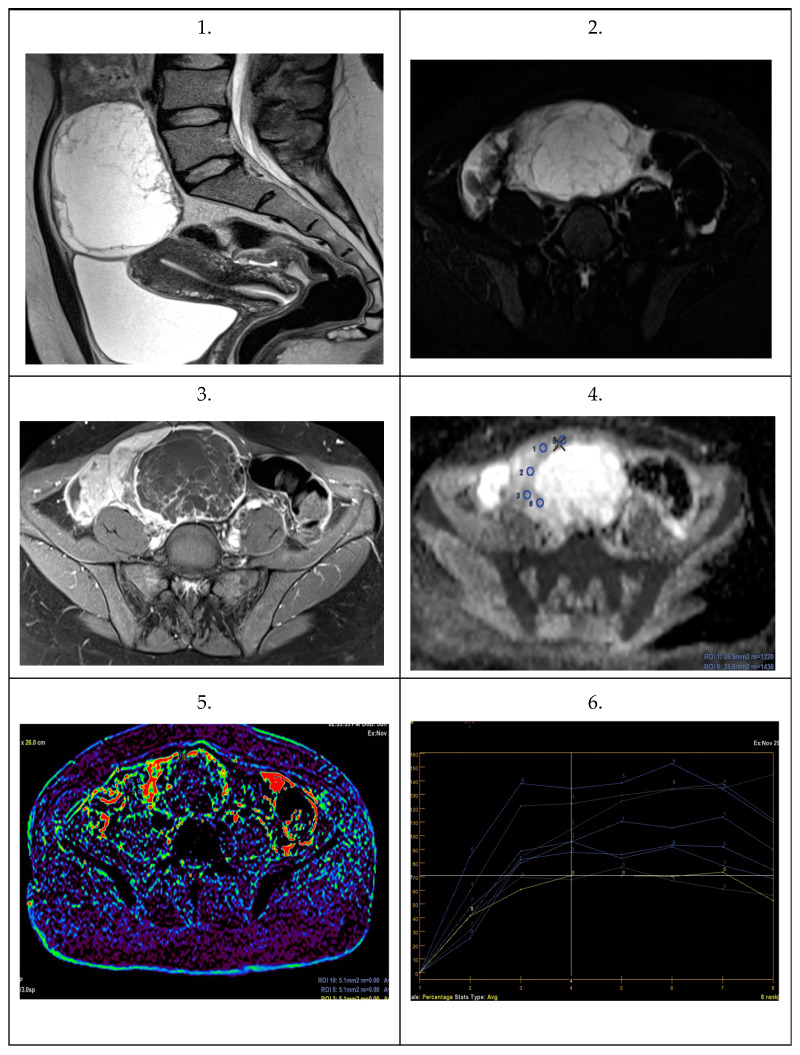
Images from 34-year-old woman with an MOC. Large primary multilocular cystic tumor. (**1**) Sagittal T2-weighted image. (**2**) T2-weighted image with fat suppression. (**3**) T1-weighted post-contrast image. (**4**) Diffusion ADC maps. (**5**) Perfusion map. (**6**) Contrast enhancement curves.

**Table 1 cancers-16-03569-t001:** MR study protocol.

Parameter	T2 TSE	T2 TSE Fat-Sat	DW EPI	T2 TIRM	Vibe 3D T1 GRE	T1 GRE (In- and Outphase)	T1 TSE Fat-Sat	T2 TSE (BLADE) Fat-Sat (SPAIR)
Repetition time [ms]	4250	2110	3800	6100	3.05	125	510	2300
Echo time [ms]	117	123	73	39	1.13	1:2.22 2:4.92	9.6	116
Flip angle [deg.]	137	150	90	150	10	70	150	150
iPAT factor	-	2	2	-	2	2	-	2
Plane	Axial, sagital, coronal	Axial	Axial	Axial	Axial	Axial	Axial, sagital, coronal	Axial, coronal
Number of signal averages	1	1	4	1	1	1	1	1
Field of view—FOV [mm]	360	360	360	360	360	360	360	360
Rectangular FOV [%]	75, 100, 100	100	75	75	75	75	75	100
Breath-hold	No	No	No	No	No	No	No	Yes
Resolution (mm)	0.7 × 0.7 × 5	1.4 × 1.4 × 5	B value: 0, 50, 500, 1000, 1500	0.9 × 0.9 × 5	1.7 × 1.3 × 3	1.3 × 1.3 × 5	0.9 × 0.9 × 5	1.4 × 1 × 4 × 6

**Table 2 cancers-16-03569-t002:** The characteristics in the group of studied patients.

Characteristic	Total	Group 1 MOC	Group 2 MOT	*p* Value
	N = 35	N = 10	N = 25	
Age (y)	56 ± 12.3	49.3 ± 12.1	58.7 ± 11.5	0.04
Stage		FIGO I—7 FIGO II—2 FIGO III—1		
Histopathology/origin		Mucinous ovarian Carcinoma—10	Colrectal cancer—22 Gastric cancer—3	
CA 125 (n: 35 U/mL) CA 19.9 (n: 37 U/mL) CEA (n: 2.5 ng.mL) median/range	35.5/10–345 28/6–378 7.0/2.0–31.8	81.5/23–345 69/10–169 2.85/2.0–3.4	32.0/10–200 23/6–378 10.1/3.0–31.8	0.01 0.4 0.0007

**Table 3 cancers-16-03569-t003:** Description of the Seidman criterion.

10 cm Criterion	Bilateral	The Seidman Criterion
≥10 cm	One-sided	MOC 82%
<10 cm	One-sided	meta 87%
≥10 cm	Two-sided	meta 95%
<10 cm	Two-sided	meta 92%

**Table 4 cancers-16-03569-t004:** The bilateral criterion, 10 cm criterion, and the Seidman criterion characteristics in concordance with histopathological postoperative results in the group of studied patients.

Parameter		Total	Primary Ovarian Cancer	Metastatic Ovarian Tumor	*p* Value
		*N* = 57	*N* = 10	*N* = 47	
Bilateral	One-sided [n (%)]	13 (23%)	10 (100%)	3 (6%)	<0.000001
	Two-sided [n (%)]	44 (77%)	0 (0%)	44 (94%)	
10 cm criterion	<10 cm [n (%)]	39 (68%)	2 (20%)	37 (79%)	0.0003
	≥10 cm [n (%)]	18 (32%)	8 (80%)	10 (21%)	
The Seidman criterion	MOC 82% [n (%)]	8 (14%)	8 (80%)	0 (0%)	<0.000001
	meta 87% [n (%)]	5 (9%)	2 (20%)	3 (6%)	
	meta 95% [n (%)]	10 (18%)	0 (0%)	10 (21%)	
	meta 92% [n (%)]	34 (60%)	0 (0%)	34 (72%)	

Test of significance, *p* value: chi-square or Fisher’s exact test.

**Table 5 cancers-16-03569-t005:** The ADC 1–4 characteristics in the group of studied female patients.

Parameter	Statistic	Total	Primary Ovarian Cancer	Metastatic Ovarian Tumor	*p* Value
		*N* = 57	*N* = 10	*N* = 47	
ADC 1	Median	851	1465.5	842	0.0001
	Q_1_–Q_3_	814–888	1430.5–1537.25	810.5–875	
ADC 2	Median	843	1434	821	0.0001
	Q_1_–Q_3_	800–890	1345.75–1643	800–872	
ADC 3	Median	846	1631.75	824	0.0001
	Q_1_–Q_3_	808–899	1400.5–1678.25	806–871.5	
ADC 4	Median	865	1442	845	0.0002
	Q_1_–Q_3_	811–998	1230.25–1627.35	809–895	

Test of significance, *p* value: the Mann-Whitney U test, Q_1_–Q_3_: the first quartile-the third quartile.

**Table 6 cancers-16-03569-t006:** The TTP and Perf. Max. En. characteristics in the group of studied female patients.

Parameter	Statistic	Total	Primary Ovarian Cancer	Metastatic Ovarian Tumor	*p* Value
		*N* = 57	*N* = 10	*N* = 47	
TTP	Median	157	410	154	0.0001
	Q_1_–Q_3_	149–171	370–465	147.5–161	
Perf.Max En.	Median	167	141	167	0.5
	Q_1_–Q_3_	143–193	135.25–198.25	145–189.5	

Test of significance, *p* value: the Mann-Whitney U test, Q_1_–Q_3_: the first quartile—the third quartile.

**Table 7 cancers-16-03569-t007:** Test of significance: Fleiss’ *kappa* for qualitative tumors assessment by mille-feuille sign, performed by 2 observers.

Category	Fleiss’ *kappa* (k)	95% CI	SE	*p* Value
Overall	0.64	0.38–0.9	0.132	0.000002
Absence	0.64	0.38–0.9	0.132	0.000002
Occurence	0.64	0.38–0.9	0.132	0.000002

k—kappa value, CI—Confidence interval, SE—standard error.

**Table 8 cancers-16-03569-t008:** Intraclass correlation of the ADC parameter performed by the two observers.

Parameter	ICC	95% CI	*p* Value
ADC	0.84	0.73–0.91	<0.000001

ICC—Intraclass correlation coefficient, CI—Confidence interval.

**Table 9 cancers-16-03569-t009:** Table with the results obtained during the analysis of ROC curves.

Parameters	Range	Sensitivity	Specificity	Cutoff Value	AUC	*p* Value
ADC	798–1733.5	98%	100%	≤912.75	0.996	0.000001
TTP	117–500	100%	100%	≤191	1	0.000001
Perf. Max. En.	120–350	83%	60%	≥143	0.57	0.475625
Mille-feuille sign	0–1	68%	80%	=1	0.74	0.017747
10 cm criterion	0–1	79%	80%	=0	0.79	0.003787
The Seidman criterion	1–4	94%	100%	≥3	0.994	0.000001

## Data Availability

The raw data supporting the conclusions of this article will be made available by the authors on request.

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
