# Peer review of "Evaluation of Selected MRI Parameters in the Differentiation of Mucinous Ovarian Carcinomas and Metastatic Ovarian Tumors"

_cancers, 2024, doi:10.3390/cancers16213569_

Round 1
Reviewer 1 Report
Comments and Suggestions for Authors
The authors reported an interesting article. However, some revisions are needed to improve the manuscript.
Specific critiques
Abstract
The article would be more helpful to clinicians if it clearly related the role of mri to the diagnosis of MOC or MOT.
Specific critiques
Title and Abstract
Title Consistency: ‘Evaluation of selected MRI parameters in the differentiation of 2
mucinous ovarian carcinomas and metastatic ovarian tumors’ The title reflects the entire scope of the paper but it’s too long.
Abstract Clarity: The abstract is somewhat repetitive and lacks clarity.
Introduction
Please improve this section,trying to be more clear. The epidemiology could use more detailed statistical data and references to support the claims made about the incidence, prevalence and treatment of the scar pregnancies.
Materials and Methods
This section should be corrected :
The exclusion criteria is not clear.
Please trying to be more clear about the type of study, number of patients, exclusion and inclusion criteria
Insert a separate paragraph where you talk about the morphological criteria without including them in the ‘MRI protocol’.
Results
I like this section and the tables are interesting and nice. However sentences are too complex and may induce confusion in the reader. There are too tables.
The table with patients characteristic can be improved and more clear.
Take table 1 of this article as an example:
Kurokawa R, Nakai Y, Gonoi W, Mori H, Tsuruga T, Makise N, Ushiku T, Abe O. Differentiation between ovarian metastasis from colorectal carcinoma and primary ovarian carcinoma: Evaluation of tumour markers and "mille-feuille sign" on computed tomography/magnetic resonance imaging. Eur J Radiol. 2020 Mar;124:108823. doi: 10.1016/j.ejrad.2020.108823. Epub 2020 Jan 9. PMID: 31935596.
Statistical Analysis
In this section you should be more cleare, It's too long and confusing.
Discussion
Rewrite the paragraph,trying to be clearer. Some sentences are too long and confused. Try to follow a schedule to make the discussion more usable.
You talked about Krukenberg's tumor without having introduced it before.
Conclusions
Summary: The conclusion is somewhat vague and does not effectively summarize the key points of the case. It should provide a clear summary of the findings and their implications.
Comments on the Quality of English Language
General Points.
Language and Formatting: The manuscript needs thorough proofreading for grammatical errors and formatting issues.
Consistency in style and terminology should be maintained throughout the paper.
Conclusions
It should provide a clear summary of the findings and their implications.
Future Research: There should be a more explicit call for future research, specifying which areas need further investigation.
Current Limitations: The limitations of current imaging techniques should be discussed more thoroughly to provide a balanced view.
Author Response
REPLY TO REVIEWER 1
Title and Abstract
Title Consistency: ‘Evaluation of selected MRI parameters in the differentiation of
mucinous and metastatic ovarian cancers’
The title reflects the entire scope of the paper but it’s too long.
Thank you for the remark, title has been changed.
Abstract Clarity: The abstract is somewhat repetitive and lacks clarity.
Thank you for the remark. The abstract has been modified and we hope it meet your expectations.
Introduction
Please improve this section, trying to be more clear. The epidemiology could use more detailed statistical data and references to support the claims made about the incidence, prevalence and treatment of the scar pregnancies.
Thank you for the remark however, we are afraid that your suggestion is not applicable to our article as the study isn’t about prevalence and treatment of the scar pregnancies. The epidemiology and prevalence data are supported by relevant articles no. 1,2,3 and 4.
Materials and Methods
This section should be corrected :
The exclusion criteria is not clear.
Thank you for the remark, we have made appropriate modifications:
Patients with coexisting second cancer or with contraindications to MRI with gadolinium contrast examination and under 18 years of age were excluded from the study.
Please trying to be more clear about the type of study, number of patients, exclusion and inclusion criteria
Thank you for the remark. The paragraph has been changed.
The inclusion criterion for the study purposes was a unilateral or bilateral, solid mass or mixed- solid/cystic mass diagnosed in CT or transvaginal ultrasound examination.
Insert a separate paragraph where you talk about the morphological criteria without including them in the ‘MRI protocol’.
Thank you, the morphological criteria have been separated from the study protocol.
Results
I like this section and the tables are interesting and nice. However sentences are too complex and may induce confusion in the reader. There are too tables.
The table with patients characteristic can be improved and more clear.
Take table 1 of this article as an example:
Kurokawa R, Nakai Y, Gonoi W, Mori H, Tsuruga T, Makise N, Ushiku T, Abe O. Differentiation between ovarian metastasis from colorectal carcinoma and primary ovarian carcinoma: Evaluation of tumour markers and "mille-feuille sign" on computed tomography/magnetic resonance imaging. Eur J Radiol. 2020 Mar;124:108823. doi: 10.1016/j.ejrad.2020.108823. Epub 2020 Jan 9. PMID: 31935596.
Thank you for the comment. According to the tables in our article, Table 2 describes the differences in the age of the Patients in analyzed groups. We have added data on the clinical characteristics of the patients to Table 2. Other parameters, such as diameter, location- referred as Seidman criteria, were analyzed in Tables 3 and 4, as in our material data is more extensive than I Kurokawa’s paper and it could not be so clear for the reader. However, we are open for the remarks.
Statistical Analysis
In this section you should be more cleare, It's too long and confusing.
Thank you for your attention, we are aware that statistics are extensive, but this is the only way we can prove the usefulness of the MRI parameters used, which will be useful in the differential diagnosis of meta-tumors and primary mucinous tumors, which are similar in morphological criteria. But if necessary, we are open to detailed comments.
Discussion
Rewrite the paragraph,trying to be clearer. Some sentences are too long and confused. Try to follow a schedule to make the discussion more usable.
Thank you for the remark, we have corrected the sentences in discussion.
You talked about Krukenberg's tumor without having introduced it before.
Thank you for your valuable comment. The use of the name Krukenberg was due to the reference to the cited work. We replaced the name with metastases of gastric cancer to the ovary.
Conclusions
Summary: The conclusion is somewhat vague and does not effectively summarize the key points of the case. It should provide a clear summary of the findings and their implications.
Thank you for your valuable comments. We hope that the revised conclusions will be appropriate this time.
- Morphological criteria in imaging studies have lower sensitivity and specificity in the differential diagnosis of ovarian tumors compared to quantitative criteria.
- Quantitative criteria such as ADC are a very good tool for differentiating primary mucinous ovarian carcinomas (MOC) and metastatic tumors (MOT). Among the analyzed DCE parameters, very good results in differentiating MOC and MOT were obtained in the TTP.
- In the preoperative differential diagnosis of ovarian tumors, the use of morphological criteria according to Seidman, extended by the assessment of changes on ADC and TTP maps, seems to be the sufficient method of differentiation.
Reviewer 2 Report
Comments and Suggestions for Authors
This paper is a prospective analysis, performed in a single institution, aimed to evaluate selected contrast-enhanced MRI parameters of patients with suspected ovarian masses diagnosed in ultrasound examination to achieve proper preoperative differentiation between mucinous ovarian carcinomas and metastatic ovarian tumors.
The Authors evaluated not only morphological criteria such as lesion size, bilateral location, presence of “mille-feuille signs,” and so-called Seidman criteria but also diffusion-weighted imaging and dynamic contrast enhancement of each lesion.
The paper is well written and the English language is appropriate and understandable.
The clinical topics are interesting. To date, most of the studies in the worldwide available literature examined morphologic parameters such as "mille-feuille signs," Seidman criteria, and so on that try to enable preoperative diagnosis of ovarian tumors for precise treatment planning and individualization of therapy.
This paper shows that dynamic MRI parameters such as apparent diffusion coefficients, time to peek, and perfusion maximum enhancement contribute to differentiating between mucinous ovarian carcinomas and metastatic ovarian tumors.
The limitations and bias of this review are reported thoroughly including the small size and the representativeness of the study sample.
The conclusions are appropriate focusing on the need for combining the morphological criteria and the dynamic MRI parameters for more precise and accurate diagnosis of ovarian tumors.
The cited references are mostly recent and relevant publications.
Specific comments:
Could the authors provide more details on clinico-pathological Characteristics of mucinous ovarian carcinomas and metastatic ovarian tumors (digestive organ cancer other than colorectal cancer, stage of tumors, and preoperative tumor markers)?
We suggest to remove the following sentences:
Such an image is supposed to be indicative of tumor metastasis from the colon to 61 the ovary. [8] Seidman et al. proposed a very simple way to differentiate MOC and MOT 62 based on tumor size and unilateral or bilateral occurrence. Primary ovarian origin is sup- 63 posed to be evidenced by unilateral occurrence and a diameter greater than 10 cm. Such 64 coincidence is supposed to give an approximate 90% probability of diagnosing MOT. [13] 65 Of course, these are very approximate criteria. (Rows: 61-66),
For morphological analysis, Seidman's criteria were used, dividing ovarian tumors according to the following criteria: occurrence of unilateral or bilateral and tumor diameter. (Rows: 105-107).
In a next step of the study, the tumors present among the patients studied were analyzed. In the group of examined patients, 22 (63%) patients suffered from bilateral (two-sided) tumors, and thus 13 patients (37%) possessed unilateral (one-sided) tumors. On that basis, 57 tumors were analyzed, which were also divided into two groups, i.e., the first group - primary ovarian cancer while the second group - metastatic ovarian tumor. (Rows: 196-200).
We suggest introducing the number 6 instead of the alphabet letter C (Row 228).
Author Response
Thank you for your comments.
Clinical characteristics of mucinous ovarian carcinomas and metastatic ovarian tumors are included in Table 2
Repeated paragraphs have been removed
Reviewer 3 Report
Comments and Suggestions for Authors
The authors present a manuscript which aims to investigate whether selected MRI parameters can be used to distinguish mucinous ovarian carcinomas and metastatic ovarian tumors. The study has been conducted properly and the manuscript is well written. However, several corrections should be made to achieve better comprehension. First, the whole manuscript should be edited in English as it contains grammatical and typographical errors. Second, the authors should kindly specify when magnetic resonance imaging (MRI) was performed (i.e. preoperatively or postoperatively) and MRI scans were interpreted (i.e. preoperatively or postoperatively, before or after histopathological examination). The authors should also determine whether the radiologists were blinded to clinical and histopathological findings or not. Third, the authors should mention about the clinical implications of their findings in a separate paragraph of the discussion part. Lastly, the authors should replace the references that were published before 2008 with newer and more up-to-date ones if possible.
Comments on the Quality of English LanguageThe authors present a manuscript which aims to investigate whether selected MRI parameters can be used to distinguish mucinous ovarian carcinomas and metastatic ovarian tumors. The study has been conducted properly and the manuscript is well written. However, several corrections should be made to achieve better comprehension. First, the whole manuscript should be edited in English as it contains grammatical and typographical errors. Second, the authors should kindly specify when magnetic resonance imaging (MRI) was performed (i.e. preoperatively or postoperatively) and MRI scans were interpreted (i.e. preoperatively or postoperatively, before or after histopathological examination). The authors should also determine whether the radiologists were blinded to clinical and histopathological findings or not. Third, the authors should mention about the clinical implications of their findings in a separate paragraph of the discussion part. Lastly, the authors should replace the references that were published before 2008 with newer and more up-to-date ones if possible.
Author Response
Thank you for your comments
- We corrected language errors.
- We specified when magnetic resonance imaging (MRI) was performed (i.e., preoperatively or postoperatively) and when the MRI scans were interpreted (i.e., preoperatively or postoperatively). The radiologists were only aware of the pelvic mass and were familiar with preoperative marker testing, which is a routine item on the referral form. (Included in the Material and Methods section)
- A paragraph on clinical implications was added
- We changed publications written before 2008 to more recent ones (items: 9,11,26). We left item 13 (Seidman et al. Am J Surg Pathol. 2003 Jul;27(7):985-93) because we are referring to it directly
Reviewer 4 Report
Comments and Suggestions for Authors
In the manuscript Cancers-3141406 the authors analyze a few MRI parameters to differentiate mucinous ovarian carcinomas and metastatic tumors. I appreciate the efforts made by the authors, but I believe there is room for improvement. I have a few suggestions and questions.
-There are several instances of repeated sentences, such as lines 49-51 on page 2, throughout the manuscript.
-Full stop is missing at the end of sentence in line 56.
-There is an extra full stop in line 72.
-In the abstract, the authors mentioned that the inclusion criteria were ultrasound examination, while in the second section, they added a CT examination. This needs clarification.
- The parameters mentioned in line 98 are listed in Table 1, not Table 2.
- ADC’s parameters are not listed in Table 1; they might be in Table 5.
- Explanations for the (in)dependent variables used in this study (section 2.3) are necessary.
- The analysis presented in Table 3 and Table 4 needs clarity. What is the main aim of the analysis? Are both parameters in the first two rows of Table 4 already included in the Seidman criterion? Further explanations are needed.
- Table C is mentioned in line 228, but it cannot be found.
- There are no explanations of figures 13-19.
- Is it possible to create a model based on the results presented here or to introduce new features in existing protocols?
- The Discussion section could be reorganized for better readability.
- It appears that similar investigations were conducted in some of the cited references, such as Ref. [32] and Ref. [33]. More information is required to explain the differences in conclusions between these studies and the investigation presented here.
- As stated by the authors, the limitation of this study is a small sample size. Hence, the authors should temper their claims and conclusions.
In the current form, the manuscript does not deserve to be published in this journal.
Comments on the Quality of English LanguageIt can be improved.
Author Response
-There are several instances of repeated sentences, such as lines 49-51 on page 2, throughout the manuscript.
Thank you for the remark, corrected.
-Full stop is missing at the end of sentence in line 56.
Thank you for the remark, corrected.
-There is an extra full stop in line 72.
Thank you for the remark, corrected.
-In the abstract, the authors mentioned that the inclusion criteria were ultrasound examination, while in the second section, they added a CT examination. This needs clarification.
Thank you for the remark, corrected, information about CT examination as a inclusion criteria was added to the abstract section.
- The parameters mentioned in line 98 are listed in Table 1, not Table 2.
Thank you for the remark, corrected.
- ADC’s parameters are not listed in Table 1; they might be in Table 5.
Thank you for the remark, corrected.
- Explanations for the (in)dependent variables used in this study (section 2.3) are necessary.
Thank you, Explanation is added to section 2.3
- The analysis presented in Table 3 and Table 4 needs clarity. What is the main aim of the analysis? Are both parameters in the first two rows of Table 4 already included in the Seidman criterion? Further explanations are needed.
Table 3 shows the percentage of the morphological Seidman criteria (diameter and bilaterality) that are met in the described 57 tumors. Table 4 shows the percentage of the origin of the tumors described earlier according to the Seidman criteria that matches the histopathological result. The same applies to the diameters of the tumors and unilaterality or bilaterality. – we have added a clarifying sentence in the title of Table 4.
- Table C is mentioned in line 228, but it cannot be found.
Thank you for the remark, it should be mentioned Table 6 instead of Table C, corrected.
- There are no explanations of figures 13-19.
Explanations for the mentioned figures have been added in the text
- Is it possible to create a model based on the results presented here or to introduce new features in existing protocols?
In this work, we wanted to propose a diagnostic model based on the initial application of Seidman's criteria and then combining them with diffusion parameters (ADC map) and basic perfusion elements (DCE) e.g. TTP and max enhancement in order to differentiate primary and metastatic ovarian tumors. Quantitative determination of DCE parameters in particular is not currently widely used.
The Discussion section could be reorganized for better readability.
Thank you for the remark, corrected. We hope that corrections improved the value of the discussion section.
- It appears that similar investigations were conducted in some of the cited references, such as Ref. [32] and Ref. [33]. More information is required to explain the differences in conclusions between these studies and the investigation presented here.
Paper 32 concerned primary ovarian cancers with differentiation into their types I and II. Type I includes: low-grade EOC clear cell carcinoma and Brenner tumor. Type II is classic high-grade EOC. Similar analyses were performed in several studies, e.g. Lindgrem et al. (Eur. J. Radiol. 2019; 115:66–73.) as well as in our 2022 Derlatka P et al. (Cancers. 2022; 14:2464). We also differentiated serous ovarian cancers from mucinous ones. The currently presented study concerns metastatic MOT tumors and primary mucinous MOC tumors, which were not analyzed in paper 32.
Position 33 presents a review paper.
There is still position 34 in which primary cancers and metastatic tumors are differentiated. However, DWI and DCE are marked qualitatively, not quantitatively. Which we emphasize in the discussion.
- As stated by the authors, the limitation of this study is a small sample size. Hence, the authors should temper their claims and conclusions.
Thank you, we have taken into account the comments in the conclusions, we are aware of the limitations, but at the same time we wanted to draw attention to the fact that measurements, especially of perfusion parameters (which indicates clear differences in the vascularization of tumors) bring significant value in differential diagnosis.
Comments on the Quality of English Language
Thank you, the translator corrected the work, but if it is still not suitable, we will proofread it again.
Round 2
Reviewer 1 Report
Comments and Suggestions for Authors
The authors improved the manuscript following all reviewers suggestions.
Author Response
Dear Reviewer, thank you for the valuable remarks.
Reviewer 3 Report
Comments and Suggestions for Authors
The authors present their manuscript which aims to evaluate selected magnetic resonance imaging parameters in the differentiation of mucinous ovarian carcinomas and metastatic ovarian tumors. I thank the authors for their valuable efforts in making corrections and revisions. I would only recommend that the second and third sentences of the last paragraph of the discussion part should be rewritten as grammatically correct and regular sentences. Otherwise the revised version of this manuscript can be accepted for publication in Cancers.
Comments on the Quality of English Language
The authors present their manuscript which aims to evaluate selected magnetic resonance imaging parameters in the differentiation of mucinous ovarian carcinomas and metastatic ovarian tumors. I thank the authors for their valuable efforts in making corrections and revisions. I would only recommend that the second and third sentences of the last paragraph of the discussion part should be rewritten as grammatically correct and regular sentences. Otherwise the revised version of this manuscript can be accepted for publication in Cancers.
Author Response
Dear Reviewer, thank you for the valuable remarks.
Second and third sentence of the last paragraph of the Discussion was modified as follows:
Firstly, main limitation was single-center nature of the study and secondly, a small number of the patients enrolled to the study.
Reviewer 4 Report
Comments and Suggestions for Authors
I am satisfied with the authors' response. The manuscript is improved. I recommend it for publication.
Author Response

(The authors gave the same response as above.)
